# Heat Transport of Electrokinetic Flow in Slit Soft Nanochannels

**DOI:** 10.3390/mi10010034

**Published:** 2019-01-07

**Authors:** Zehua Wang, Yongjun Jian

**Affiliations:** School of Mathematical Science, Inner Mongolia University, Hohhot, Inner Mongolia 010021, China; wangzehua19931226@163.com

**Keywords:** slit soft nanochannels, streaming potential, thermal transport, entropy generation

## Abstract

Soft nanochannels are defined as nanochannels with a polyelectrolyte layer (PEL) on the rigid walls. In the present study, the thermal transport properties of the fluids through slit soft nanochannels are investigated under the combined influences of pressure-driven and streaming potential. Based on the analytical solutions of electric potential and velocity distributions, a dimensionless temperature of electrolyte solution in soft nanochannels is obtained by resolving the energy equation. Then, a finite difference method is used to compute the energy equation and test the validity of the analytical solution. Results show that the temperature increases with the decrease of dimensionless velocity and the heat transfer rate for rigid nanochannel are higher than that for the soft one. Moreover, we find the total entropy generation decreases with the increases of the ratio *K_λ_* of the electrical double layer (EDL) thickness in PEL to the EDL thickness on the solid wall.

## 1. Introduction

In recent years, microfluidic devices, including micro electro mechanical systems (MEMS), have been widely used in biology, chemistry, medicine, and engineering for chemical separation and thermal management of microelectronic systems [1,2,3,4,5,6,7,8]. There are many advantages of these devices, such as relatively low costs, light weights, high transport coefficients, small heat loss, etc. Currently, there are numerous types of technologies for driving and controlling microfluidics. In general, the fluid flow in nanochannels is driven by pressure, surface tension, external electric field, external magnetic field and high frequency sound wave [9,10]. Compared with other driven mechanisms, pressure driven flow in nanochannels has many applications [11,12,13] and is often utilized to obtain streaming potential. When the electrolyte solution comes into contact with the wall of nanochannels, an electric double layer (EDL) is formed due to the charge exchange between the wall surface and the electrolyte. A streaming current is produced along a flowing direction within the EDL when an electrolyte solution flows in channels under a pressure gradient. When the electrolyte solution is flowing, the net charge is gathered at downstream nanochannels. As a result, the downstream potential in nanochannels is higher than the one of the upstream. Finally, this induces a steady-state electric field, called the streaming potential, opposite to the original direction of flow. Relevant studies have indicated that, through the above process, the kinetic energy of the pressure-driven transport and chemical energy of the EDL can be converted into electrical energy [14,15,16,17,18,19,20,21]. The streaming potential phenomenon is considered as a reciprocal phenomenon of electroosmosis which represents the movement of electrolyte solution relative to a stationary charged surface due to an applied electric field [22].

Traditionally, nanochannels with a polyelectrolyte layer (PEL) on the walls of rigid nanochannels are labeled as soft nanochannels [23]. The PEL is assumed to contain a particular kind of ion and fixed charge density of ions. The bulk electrolyte ions can penetrate into the PEL. Therefore, the PEL electrolyte interface acts as a semi-penetrable membrane [24]. Meanwhile, there is an additional drag force on fluid flow within the PEL, similar to the flowing process of electrolyte solution through porous media. Some electrokinetic theories and phenomena in soft channels have been studied [25,26,27,28,29]. Chanda et al. [30] studied the streaming potential and electroviscous effects in soft nanochannels. They obtained the analytical solutions of the electrostatic potential, velocity and streaming potential.

Due to the need for higher heat transfer rates and lower entropy generation in industrial applications, the thermal behavior in nanochannels has attracted the attention of a host of researchers. Matin [31] analyzed the thermal transport of electrolytes by imposing pressure gradient in rigid nanoslit. He obtained the analytical solution of temperature and found the presence of electroviscous effects will dramatically boost the heat transfer rate. The entropy production is defined as irreversible behaviors of fluid systems. In recent years, some efficient systems of fluid have been designed to decrease useless energy, which can lead to the irreversibility of fluids due to the influences of heat transfer and friction dissipation. Minimization concept of entropy generation has drawn much attention in thermal engineering. 

From the literature investigation, the problem of the heat transfer in nanochannels has been studied by many researchers, but there are few studies regarding the heat transfer in soft nanochannels. Meanwhile, the heat transfer in soft nanochannels is also very significant. Therefore, the motivation in this paper is to study the heat transfer for soft nanochannels and to compare the related results with rigid nanochannels. Based upon the acquired analytical solutions of the electric potential, velocity and streaming potential [30], the entropy generation of electrokinetic flow through slit soft nanochannels will be investigated in this paper. Then, in the case of thermally fully developed flow with constant wall heat flux, we deduce the analytical solution of temperature and verify it by using a finite difference method. Finally, this paper reveals that an enhancement of the viscous dissipation effect will lead to the diminution of the heat transfer rate in soft nanochannels and the heat transfer rate for rigid nanochannels is higher than that for soft nanochannels. The results obtained in this paper may provide theoretical guidance for the equipment design of the heat transfer and energy utilization in industrial applications. The applications of this nanoscale heat transfer analysis are concerned with the thermal performance of the nanoscale systems, for example, biological cell membranes for medical science [32].

## 2. Mathematical Modeling

We consider thermally fully developed flow through slit soft nanochannels with wall spacing 2*h^*^* under the influences of imposed pressure gradient –*dp^*^/dx*^*^. Soft nanochannels are formed by grafting positively charged a polyelectrolyte layer (PEL) thickness *d^*^* (*d^*^< h^*^*) at the inner wall surface of rigid nanochannels. The bottom and upper walls of the channel are featured by the same constant charge density. As shown in Figure 1, the two-dimensional coordinate system is established at the centerline of the soft nanochannels. The flow is assumed to be steady, hydrodynamically and thermally fully developed. Also, it is assumed that the wall heat flux *q_w_^*^* is uniform and constant. Based on its symmetry, we study only the bottom half of soft nanochannels (i.e., −*h^*^* ≤ *y^*^* ≤ 0) for the analysis and its flowing direction is only along the positive *x^*^*-axis. Due to the presence of the PEL, the temperature distribution and the entropy generation are discussed inside and outside the PEL, respectively.

## 3. The Temperature Distribution and Entropy Generation Analysis

Prior to analyzing the temperature field in soft nanochannels, we firstly need to derive the electrostatic potential, velocity and streaming potential, which have been given analytically by solving the linearized Poisson-Boltzmann equation (valid for low electric potential), the Cauchy momentum equation and the net ionic current equilibrium equation in Ref. [30]. For the brevity, we no longer list these results here. In this paper, the relative permittivity *ε_r_^*^*, the density of fluid *ρ^*^*, the specific heat of the fluid *C_p_^*^*, the thermal conductivity of the fluid *k^*^*, the electrical resistivity of the liquid *σ^*^* and the dynamic viscosity of the electrolyte *μ^*^* are taken the same values in the regions outside and inside the PEL [33,34,35] by assuming a low chain grafting density. The energy equations can be presented as
(1)$$\rho^{*}C_{p}^{*}(\frac{\partial T^{*}}{\partial t^{*}}+\vec{u^{*}}\cdot \nabla T^{*})=k^{*}\nabla^{2}T^{*}+\sigma^{*}E_{S}{}^{*2}+\mu^{*}\Phi^{*},\;(-h^{*}+d^{*}\leq y^{*}\leq 0)$$
(2)$$\rho^{*}C_{p}^{*}(\frac{\partial T^{*}}{\partial t^{*}}+\vec{u^{*}}\cdot \nabla T^{*})=k^{*}\nabla^{2}T^{*}+\sigma^{*}E_{S}{}^{*2}+\mu^{*}\Phi^{*}+\mu_{c}^{*}u^{*2},\;(-h^{*}\leq y^{*}<-h^{*}+d^{*})$$
where *T^*^* is the local temperature of the fluid, *μ_c_^*^* is the (per unit volume) drag coefficient and Φ^*^ is the viscous dissipation. The right hand side of the energy equations represents heat dissipation, volumetric Joule heating and viscous dissipation. It should be mentioned that the viscous dissipation within the PEL of Equation (2) is determined by the viscous flow of the electrolyte and the extra drag force of these special ions. Owing to the uniform flow in *x*^*^ direction and the steady temperature distribution assumption, we have Φ^*^ = (*du^*^/dy^*^*)^2^ and *∂T^*^/∂t^*^* = 0. Since heat is added or removed from the fluid, it follows that its local temperature varies with distance *x*^*^ along the channel. However, we can define a fully developed dimensionless temperature profile which has a single distribution in the vertical direction at all location *x*^*^, i.e.,
(3)$$\frac{\partial }{\partial x^{*}}[\frac{T_{w}^{*}(x^{*})-T^{*}(x^{*},y^{*})}{T_{w}^{*}(x^{*})-T_{m}^{*}(x^{*})}]=0$$
where *T_w_*^*^(*x^*^*) is local wall and *T_m_*^*^(*x^*^*) is the velocity-weighted average temperature, determined as $T_{m}^{*}=\int_{0}^{h^{*}}u^{*}(y^{*})T^{*}(y^{*})dy^{*}/\int_{0}^{h^{*}}u^{*}(y^{*})dy^{*}$. Application of Newton’s law of cooling gives *q_w_^*^*= *h*′^*^ [*T*^*^*_w_*(*x^*^*) − *T*^*^*_m_*(*x^*^*)], where *h*′^*^ is the convective heat transfer coefficient. In the case of a constant wall heat flux, i.e., *q_w_^*^* and *h*^′*^ are constant, we can get *T*^*^*_w_*(*x^*^*) − *T*^*^*_m_*(*x^*^*) = const., i.e., *dT*^*^*_w_*(*x^*^*)/*dx ^*^* = d*T*^*^*_m_*(*x^*^*)/*dx^*^*. Substituting *dT*^*^*_w_*(*x^*^*)/*dx^*^* = d*T*^*^*_m_*(*x^*^*)/*dx^*^* into Equation (3), we have
(4)$$\frac{\partial T^{*}(x^{*},y^{*})}{\partial x^{*}}=\frac{dT_{m}^{*}(x^{*})}{dx^{*}}=\frac{dT_{w}^{*}(x^{*})}{dx^{*}}=\mathrm{const},\;\frac{\partial^{2}T^{*}(x^{*},y^{*})}{\partial x^{*2}}=0\;$$

Under these assumptions, the energy governing equations (Equation (1) and (2)) becomes
(5)$$\rho^{*}C_{p}^{*}u^{*}\frac{dT_{m}^{*}}{dx^{*}}=k^{*}(\frac{d^{2}T^{*}}{dy^{*2}})+\sigma^{*}E_{S}{}^{*2}+\mu^{*}{\left(\frac{du^{*}}{dy^{*}}\right)}^{2},\;(-h^{*}+d^{*}\leq y^{*}\leq 0)$$
(6)$$\rho^{*}C_{p}^{*}u^{*}\frac{dT_{m}^{*}}{dx^{*}}=k^{*}(\frac{d^{2}T^{*}}{dy^{*2}})+\sigma^{*}E_{S}{}^{*2}+\mu^{*}{\left(\frac{du^{*}}{dy^{*}}\right)}^{2}+\mu_{c}^{*}u^{*2},\;(-h^{*}\leq y^{*}<-h^{*}+d^{*})$$

The boundary conditions of Equations (5) and (6) are
(7)$$\frac{dT^{*}}{dy^{*}}{|}_{y^{*}=0}=0,\;\frac{dT^{*}}{dy^{*}}{|}_{y^{*}={\left(-h^{*}+d^{*}\right)}^{-}}=\frac{dT^{*}}{dy^{*}}{|}_{y^{*}={\left(-h^{*}+d^{*}\right)}^{+}}$$
(8)$$T^{*}{|}_{y^{*}={\left(-h*+d^{*}\right)}^{-}}=T^{*}{|}_{y^{*}={\left(-h^{*}+d^{*}\right)}^{+}},\;T^{*}{|}_{y^{*}=-h^{*}}=T_{w}^{*}$$

In physics, the boundary conditions in Equation (7) respectively denote that the temperature is symmetrical about the *y**-axis and the heat flux is continuous at the interface of the PEL and the electrolyte solution. Further, Equation (8) respectively represents that the temperature is continuous at the interface of the PEL and the electrolyte solution and the wall heat flux is constant. Non-dimensional parameters and variables of electrical potential and velocity have been used in Ref. [30]. Furthermore, an overall energy balance for an elemental control volume on a length of duct *dx*^*^ along the *x*^*^-axis is applied, integrating the Equations (5) and (6) in two regions respectively and we have
(9)$$\begin{array}{l} (-h^{*}+d^{*})\rho^{*}C_{p}^{*}u_{m1}^{*}dT_{m}^{*}+d^{*}\rho^{*}C_{p}^{*}u_{m2}^{*}dT_{m}^{*}=\\ q_{w}^{*}dx^{*}+h^{*}\sigma^{*}E_{S}^{*2}dx^{*}+\mu^{*}\int_{-h^{*}+d^{*}}^{0}{\left(\frac{du^{*}}{dy^{*}}\right)}^{2}dy^{*}dx^{*}+\mu^{*}\int_{-h^{*}}^{-h^{*}+d^{*}}{\left(\frac{du^{*}}{dy^{*}}\right)}^{2}dy^{*}dx^{*}+\int_{-h^{*}}^{-h^{*}+d^{*}}\mu_{c}^{*}u^{*2}dy^{*}dx^{*} \end{array}$$

Rearranging Equations (9), the axial bulk temperature gradient in the thermally fully developed situation has yielded as
(10)$$\frac{dT_{m}^{*}}{dx^{*}}=\frac{q_{w}^{*}+h^{*}\sigma^{*}E_{S}{}^{*}+\mu^{*}u_{p,0}^{*2}\beta_{3}/h^{*}+\mu^{*}u_{p,0}^{*2}\beta_{4}/h^{*}+h^{*}\mu_{c}^{*}u_{p,0}^{*2}\beta_{5}}{(h^{*}-d^{*})\rho^{*}C_{p}^{*}u_{m1}^{*}+d^{*}\rho^{*}C_{p}^{*}u_{m2}^{*}}$$
where *u_p_*_,0_*^*^* is pressure-driven velocity scale, *u_m_*_1_*^*^*, *u_m_*_2_*^*^* represent the axial mean velocity within and without PEL, respectively. They can respectively be calculated by integration of *u*^*^(*y*^*^) across the section of the soft nanochannels
(11)$$u_{m1}^{*}=\frac{1}{h^{*}-d^{*}}\int_{-h^{*}+d^{*}}^{0}u^{*}(y^{*})dy^{*}=\frac{u_{p,0}^{*}}{1-d}\int_{-1+d}^{0}u(y)dy=\frac{u_{p,0}^{*}}{1-d}\beta_{1}$$
(12)$$u_{m2}^{*}=\frac{1}{d^{*}}\int_{-h^{*}}^{-h^{*}+d^{*}}u^{*}(y^{*})dy^{*}=\frac{u_{p,0}^{*}}{d}\int_{-1}^{-1+d}u(y)dy=\frac{u_{p,0}^{*}}{d}\beta_{2}$$
(13)$$\begin{array}{l} \beta_{1}=\int_{-1+d}^{0}udy,\;\beta_{2}=\int_{-1}^{-1+d}udy,\;\beta_{3}={\int_{-1+d}^{0}\left(\frac{du}{dy}\right)}^{2}dy,\;\beta_{4}={\int_{-1}^{-1+d}\left(\frac{du}{dy}\right)}^{2}dy\\ \beta_{5}=\int_{-1}^{-1+d}u^{2}dy \end{array}$$

Based on the analytical solutions of dimensionless velocity in Ref. [30], *β*_1_, *β*_2_, *β*_3_, *β*_4_ and *β*_5_ can be calculated analytically. However, they are not listed here due to their long expressions. Introducing the following dimensionless temperature
(14)$$\theta \left(y\right)=\frac{T^{*}(y^{*})-T_{w}^{*}}{q_{w}^{*}h^{*}/k^{*}}$$
the dimensionless energy equations and boundary conditions can be presented as:(15)$$\frac{d^{2}\theta }{dy^{2}}=\frac{u(1+JE_{0}^{2}+Br\beta_{3}+Br\alpha^{2}\beta_{5})}{\beta_{1}+\beta_{2}}-JE_{0}^{2}-Br{\left(\frac{du}{dy}\right)}^{2},\;(-1+d\leq y\leq 0)$$
(16)$$\frac{d^{2}\theta }{dy^{2}}=\frac{u(1+JE_{0}^{2}+Br\beta_{3}+Br\beta_{4}+Br\alpha^{2}\beta_{5})}{\beta_{1}+\beta_{2}}-JE_{0}^{2}-Br{\left(\frac{du}{dy}\right)}^{2}-Br\alpha^{2}u^{2},\;(-1\leq y<-1+d)$$
(17)$$\frac{d\theta }{dy}{|}_{y=0}=0,\;\frac{d\theta }{dy}{|}_{y={\left(-1+d\right)}^{-}}=\frac{d\theta }{dy}{|}_{y={\left(-1+d\right)}^{+}}$$
(18)$$\theta {|}_{y={\left(-1+d\right)}^{-}}=\theta {|}_{y={\left(-1+d\right)}^{+}},\;\theta {|}_{y=-1}=0$$

Physically, the parameter *J* (i.e., *J* = *h^*^σ^*^E_S_^*^*^2^/*q_w_^*^*) commonly called the dimensionless Joule heating parameters represents the ratio of Joule heating to the wall heat flux of soft nanochannels, and the parameter *Br*, termed the Brinkmann number, *Br* = *μ^*^u_p_*_,0_*^*^*^2^/ *q_w_^*^h^*^*, denotes the ratio of heat generated by viscous dissipation to heat transport by molecular conduction. Then using the dimensionless velocity distribution in the dimensionless energy equations (15) and (16), the analytical solution of non-dimensional temperature can be obtained by integrating twice and applying the boundary conditions (15–18)
(19)$$\begin{array}{l} \theta =(\frac{A_{11}}{24}-\frac{Br}{12})y^{4}+\lambda^{2}A_{11}A_{1}\cosh(\frac{y}{\lambda })+\frac{A_{11}A_{2}-J}{2}y^{2}-2BrA_{1}[\lambda y\sinh(\frac{y}{\lambda })-2\lambda^{2}\cosh(\frac{y}{\lambda })]\\ -\frac{BrA_{1}^{2}}{\lambda^{2}}[\frac{\lambda^{2}}{4}\cosh(\frac{2y}{\lambda })-\frac{y^{2}}{2}]+C_{1}^{\prime },\;(-1+d\leq y\leq 0) \end{array}$$
(20)$$\begin{array}{l} \theta =\frac{D_{1}e^{\alpha y}}{\alpha^{2}}-\frac{D_{2}e^{-\alpha y}}{\alpha^{2}}+\lambda^{2}D_{3}\cosh(\frac{y}{\lambda })+\lambda^{2}D_{4}\sinh(\frac{y}{\lambda })+\frac{D_{5}e^{2\alpha y}}{4\alpha^{2}}+\frac{\lambda^{2}D_{7}}{4}\cosh(\frac{2y}{\lambda })+\frac{\lambda^{2}D_{8}}{4}\sinh(\frac{2y}{\lambda })\\ +\frac{(D_{9}\alpha \lambda^{2}-D_{10}\lambda )\alpha \lambda^{2}}{{\left(\lambda^{2}\alpha^{2}-1\right)}^{2}}[\sinh(\frac{y}{\lambda })e^{\alpha y}-\frac{e^{\alpha y}}{\lambda \alpha }\cosh(\frac{y}{\lambda })]+\frac{(D_{10}\lambda^{2}\alpha -D_{9}\lambda )\lambda^{2}\alpha }{{\left(\lambda^{2}\alpha^{2}-1\right)}^{2}}[\cosh(\frac{y}{\lambda })e^{\alpha y}-\frac{e^{\alpha y}}{\alpha \lambda }\sinh(\frac{y}{\lambda })]\\ +\frac{(D_{11}\lambda^{2}\alpha -D_{12}\lambda )\lambda^{2}\alpha }{{\left(\lambda^{2}\alpha^{2}-1\right)}^{2}}[\sinh(\frac{y}{\lambda })e^{-\alpha y}+\frac{e^{-\alpha y}}{\alpha \lambda }\cosh(\frac{y}{\lambda })]+\frac{(D_{11}\lambda +D_{12}\lambda^{2}\alpha )\lambda^{2}\alpha }{{\left(\lambda^{2}\alpha^{2}-1\right)}^{2}}[\cosh(\frac{y}{\lambda })e^{-\alpha y}-\frac{e^{-\alpha y}}{\alpha \lambda }\sinh(\frac{y}{\lambda })]\\ +\frac{D_{6}e^{-2\alpha y}}{4\alpha^{2}}+\frac{D_{13}y^{2}}{2}+C_{2}^{\prime }y+C_{3}^{\prime },\;(-1\leq y<-1+d) \end{array}$$

Relevant coefficients are expressed in Appendix A. According to the calculated velocity and temperature distributions, the non-dimensional bulk temperature can be defined as
(21)$$\theta_{m}=\int_{0}^{1}u(y)\theta (y)dy/\int_{0}^{1}u(y)dy=k^{*}(T_{m}^{*}-T_{w}^{*})/q_{w}^{*}h^{*}$$

Furthermore, we can define the significant heat transfer rate regarded as the Nusselt number *Nu*, which is written as
(22)$$Nu=\frac{2h^{*}q_{w}^{*}}{k^{*}(T_{w}^{*}-T_{m}^{*})}$$

Substituting Equation (21) into Equation (22), *Nu* can be finally presented as
(23)$$Nu=-\frac{2}{\theta_{m}}$$

Finally, based on the analytical solutions of velocity and temperature distributions in Equations (19) and (20), *Nu* can be calculated by Equation (23).

For prescribed velocity and temperature distributions, we can define the entropy generation rate in the soft nanochannels. Based on the theory of Bejan [36], the entropy generation rate of the volume for the current problems can be defined as
(24)$$S_{G,L}^{*}=S_{G,H}^{*}+S_{G,J}^{*}+S_{G,V}^{*}$$
where *S_G,L_^*^* represents the volumetric entropy generation rate (per unit volume), and the terms on the right hand side of equation denote the irreversibility of local volumetric entropy generation rate by virtue of heat diffusion, Joule heating and viscous friction of the fluids, respectively. They can be written as
(25)$$S_{G,H}^{*}=\frac{k^{*}}{T^{*2}}{\left(\frac{dT^{*}}{dy^{*}}\right)}^{2},\;S_{G,J}^{*}=\frac{\sigma^{*}}{T^{*}}E_{S}^{*2},\;S_{G,V}^{*}=\frac{\mu^{*}}{T^{*}}{\left(\frac{du^{*}}{dy^{*}}\right)}^{2}$$

Using the characteristic entropy transfer rate *k^*^*/*h^*^*^2^ to non-dimensional the volumetric entropy generation, and its dimensionless form is
(26)$$S_{G}=S_{H}+S_{J}+S_{V}$$
where
(27)$$S_{H}=\frac{1}{{\left(\theta +\Theta \right)}^{2}}{\left(\frac{d\theta }{dy}\right)}^{2},\;S_{J}=S_{1}\frac{1}{\theta +\Theta },\;S_{V}=Br\frac{1}{\left(\theta +\Theta \right)}{\left(\frac{du}{dy}\right)}^{2}$$
here *Θ* = *k^*^T_w_^*^*/*h^*^q_w_^*^* is a nondimensional wall temperature, which is set to a constant in the following computation. The dimensionless global entropy generation rate can be calculated by integrating Equation (26) in soft nanochannels, i.e.,
(28)$$S_{total}=\int_{-1}^{-1+d}S_{G}dy+\int_{-1+d}^{0}S_{G}dy$$

## 4. Results and Discussions

Chanda et al. [30] have discussed the electrostatic potential and streaming potential in soft nanochannels and found that the electrostatic potential increases as the growth of *d* or the decreases of the ratio *K_λ_* of the electrical double layer (EDL) thickness in PEL to the EDL thickness on the solid wall. Accordingly, we only discuss the variations of the dimensionless velocity, temperature and entropy generation in soft nanochannels based on the above obtained analytical solutions. It needs to discuss the permissible ranges of relevant parameters in analyzing the electrokinetic flow and heat transfer of fluids. The classical parameter values are determined as follow: the permittivity of free space *ε*_0_*^*^* approximately equals to 8.854 × 10^−12^ C^2^/N·m^2^, the absolute temperature *T_a_^*^* equals to 298 K, *e^*^* equals to 1.602 × 10^−19^ C, the reference Helmholtz-Smoluchowski electroosmotic velocity *u_e_*_,0_*^*^* is set to be 10^−4^ m s^−1^, and the value of *k_B_^*^* (Boltzmann constant) is 1.381 × 10^−23^ J·K^−1^. The height of the soft nanochannels is *h^*^* > 20nm, the coefficient of viscosity of the liquid *μ^*^*~10^−3^–1.5 × 10^−3^ kg/(m s), the Brinkmann number is *Br*~0–0.1 and the dimensionless Joule heating parameters *J*~0–10 [37,38]. In this paper, *d* should satisfy the range of *d* ≤ *λ* [24,30]. Furthermore, in order to satisfy the linearized approximation of the potential distribution, the *K_λ_* should satisfy *K_λ_* ≥ 1 [24,30].

Due to no discussions of the variations of the velocities with related parameters in Chanda et al. [30], we will depict the change of the dimensionless velocity given by Chanda et al. [30] in the soft nanochannels with *y* for different values *d* and *K_λ_* in Figure 2. It can be found from Figure 2 that the dimensionless velocity of the electrolyte solution increases with *y*, which indicates the velocity attain the largest value at the centerline of soft nanochannels. The velocity inside the PEL is less than the one outside the PEL. In addition, we can also see from Figure 2a that the velocity throughout the entire soft nanochannels decreases with the increase of *d*. As a matter of fact, an increase of the *d* gives rise to an increase of the electric potential, i.e., the increasing ionic concentration, which further causes an enhancement of the drag force and finally leads to a decrease of the velocity. It is observed From Figure 2b that the velocity has an increasing trend as an augment of *K_λ_*. From a physical point of view, this is because an increase of *K_λ_* gives rise to a decrease of the electric potential throughout the entire soft nanochannels, which further leads to a decrease of the streaming potential, finally causing a rise of the velocity. 

Figure 3 gives the comparison of the analytical solutions of temperature with numerical solutions obtained by using a finite difference method. The detailed algorithm can be found in Appendix B. It is shown that the analytical solutions agree well with the numerical results. It can be found from the expression of dimensionless temperature (14) that *θ* is related to the temperature difference (*T*^*^*_w_*−*T*^*^) between the wall of soft nanochannels and electrolyte solution. Hence, the magnitude of dimensionless temperature *θ* becomes larger with the temperature difference. From Figure 3, the magnitude of dimensionless temperature *θ* has an increasing trend far away from the wall. It can also be observed from Figure 3a that the dimensionless temperature reduces as the increasing value of *K_λ_*. This can be interpreted by using energy balance. As a matter of fact, the velocity grows with *K_λ_*, which can enhance thermal energy further. This is because the thermal energy is transferred from the wall of soft nanochannels to the fluid by the flowing fluid, which eventually brings about rising in local temperature of the fluid and a decrease of the wall temperature, i.e., the decreasing temperature difference. Figure 3b reflects that the dimensionless temperature curve falls as the increasing *d*. Similar to the above analysis, an increase of *d* can result in a decline of the velocity, which further weakens the thermal energy and eventually leads to an increase of the dimensionless temperature.

Figure 4 describes the variation of the heat transfer rate *Nu* with the Brinkmann number *Br*. It can be seen from Figure 4 that *Nu* decreases with increasing *Br*. The *Br* actually represents the degree of the viscous dissipation effect physically. The wall temperature is larger than the main temperature as the growing *Br*, i.e., the value of (*T*^*^*_w_*−*T*^*^) will become larger. According to the definition of the constant wall heat flux *q_w_^*^ = h*′^*^ (*T*^*^*_w_*−*T*^*^), the convective heat transfer coefficient *h*′^*^ naturally falls. Hence, *Nu* will finally reduce. Figure 4 (a) demonstrates the heat transfer rate become larger with *K_λ_*. The reason for this is that increasing *K_λ_* will bring about a decrease of the nondimensional temperature, i.e., the value of (*T*^*^*_w_*−*T*^*^) will become smaller. Thus, the heat transfer rate enhances.

Figure 4b shows the variation of the heat transfer rate *Nu* with the *Br* for the soft and the rigid nanochannels for the same set of parameters. In order to compare the heat transfer rate in soft and rigid nanochannels, we need to calculate the dimensionless temperature for the rigid nanochannels. The dimensionless velocity for the soft nanochannels is obtained from Ref. [30], whereas that for the rigid nanochannels can be expressed as *u* = *y*^2^/2 + *A_r_*_2_cos*h*(*y/λ*) − ½ − *u_r_E_S_ζ*, *ζ* = *ψ*_0_. Based on the analytical solution of dimensionless electrostatic potential (in Ref. [30]), velocity and streaming potential (in Ref. [30]) for the rigid nanochannels, the dimensionless temperature for the rigid nanochannels can be expressed as
(29)$$\begin{array}{l} \theta =(\frac{A_{r1}}{24}-\frac{Br}{12})y^{4}+\frac{y^{2}}{2}(-\frac{A_{r1}}{2}-u_{r}E_{S}\zeta A_{r1}-JE_{0}^{2}+\frac{BrA_{r2}^{2}}{2\lambda^{2}})-2BrA_{r2}\lambda y\sinh(y/\lambda )-\frac{BrA_{r2}^{2}}{8}\cosh(2y/\lambda )\\ +(4BrA_{r2}\lambda^{2}+A_{r1}A_{r2}\lambda^{2})\cosh(y/\lambda )+D_{2}^{\prime } \end{array}$$

Relevant coefficients are expressed in Appendix A. Then, we can acquire *Nu* by calculating. It is clearly shown in Figure 4b that the heat transfer rate for rigid nanochannels is higher than that for soft nanochannels and the heat transfer rate drops with increasing *d*. This is due to an enhancement of the nondimensional temperature with the increase of *d*. 

The effects of the EDL thickness *λ* and Joule heating parameters *J* on Nu have been severally shown in Figure 5. The influence of *λ* on the streaming potential has been described [30]. When *λ* increases, the streaming potential will become larger and velocity will become small, which results in a larger value of the nondimensional temperature. Based upon the discussion of Figure 4, the larger nondimensional temperature yields small *h*′^*^. Thus, the heat transfer rate drops as an increase of *λ* (see Figure 5a). Meanwhile, it is seen from Figure 5b that the heat transfer rate decreases with enhancing *J*. The enhancement in *J* weaken the fluidic heat transfer effect under the effect of the imposed pressure gradient. Moreover, the heat transfer rate does not vary with the change of *J* in the case of *d* = 0. This is because the streaming potential equals to zero in this situation. So, the Joule heating effect is ignored, leading to a constant *Nu*. 

Figure 6 provides the change of the local entropy generation for different values of *K_λ_*. We can observe from Figure 6 that the entropy generation grows from the center axis to the wall of soft nanochannels and has a minimum at the centerline of soft nanochannels. In addition, the entropy generation rate drops as growth of *K_λ_*. In other words, growth of *K_λ_* can reduce the irreversibility of the fluids.

The influence of Brinkmann number *Br* on the total entropy generation rate throughout entire soft nanochannels has been depicted in Figure 7. It can be seen from Figure 7 that the total entropy generation rate obviously increases with *Br*. That is to say, an enhancement of viscous dissipation effect leads to the enhancement of the fluidic irreversibility. In this case, the viscous dissipation plays a crucial role. 

## 5. Conclusions

In this work, we studied thermally fully developed thermal transport and entropy generation of electrokinetic flow under the influences of imposed pressure gradient throughout the entire slit soft nanochannels. Consequently, we deduced the analytical and numerical solution of the temperature and discuss the effects of *d* and *K_λ_* on the fluidic flow and heat transfer. The following conclusions can be drawn. Firstly, when *d* rises or *K_λ_* decreases, the nondimensional temperature increase and the heat transfer rate *Nu* decreases. Secondly, the heat transfer rate declines as the enhancement of *J* or *λ* and the heat transfer rate for rigid nanochannels is higher than that for soft nanochannels. The local entropy generation grows from the centerline to the wall of soft nanochannels. Additionally, the total entropy generation grows significantly with the *Br*. And the total entropy generation reduces as the increase in *K_λ_*, which weakens the thermal irreversibility in the microfluidic system.

## Figures and Tables

**Figure 1 micromachines-10-00034-f001:**
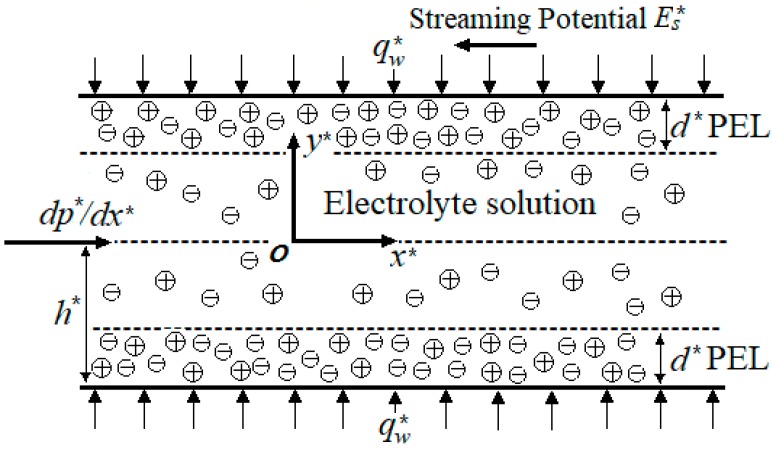
Schematic of slit soft nanochannels.

**Figure 2 micromachines-10-00034-f002:**
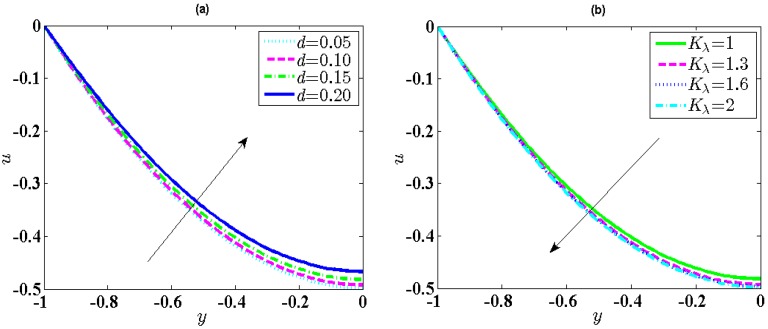
Distribution of the velocity when *λ* = 0.25, *α* = 1, *u_r_* = 0.1, *R* = 1. (**a**) Influence of *d* (0.05, 0.10, 0.15, 0.20) when *K_λ_* = 1; (**b**) Influence of *K_λ_* (1, 1.3, 1.6, 2) when *d* = 0.15.

**Figure 3 micromachines-10-00034-f003:**
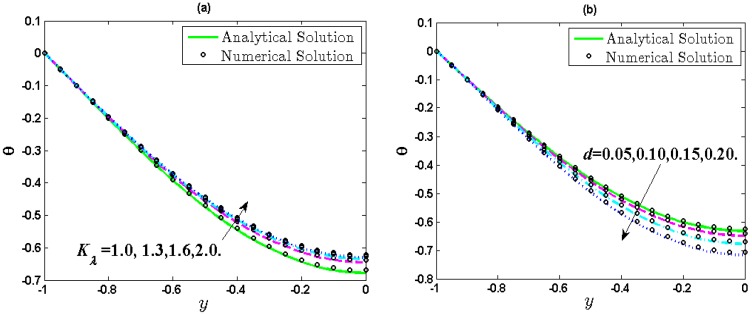
Distribution of temperature when *λ* = 0.25, *R* = 1, *α* = 1, *u_r_* = 0.1, *Br* = 0.01, *J* = 5. (**a**) Influence of *K_λ_* (1, 1.3, 1.6, 2) when *d* = 0.15; (**b**) Influence of *d* (0.05, 0.10, 0.15, 0.20) when *K_λ_* = 1.

**Figure 4 micromachines-10-00034-f004:**
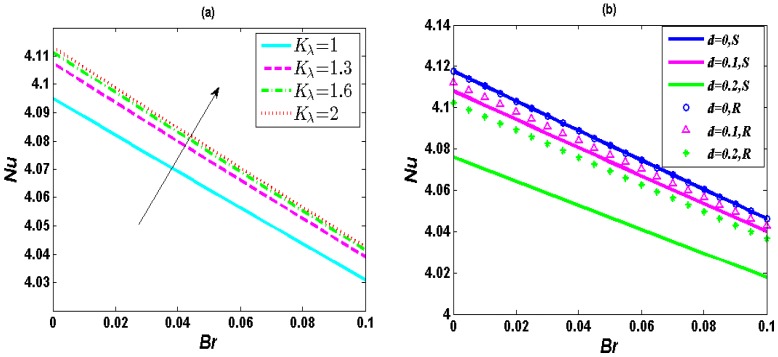
Distribution of *Nu* with *Br* (0–0.1) when *λ* = 0.25, *R* = 1, *α* = 1, *u_r_* = 1, *J* = 5. (**a**) Influence of *K_λ_* (1, 1.3, 1.6, 2) when *d* = 0.15; (**b**) Distribution of *Nu* with *Br* (0–0.1) in the channel bottom half for the soft and the rigid nanochannels for different values of the *d* for *λ* = 0.25, *R* = 1, *α* = 1, *u_r_* = 1, *J* = 5, *K_λ_* = 1.

**Figure 5 micromachines-10-00034-f005:**
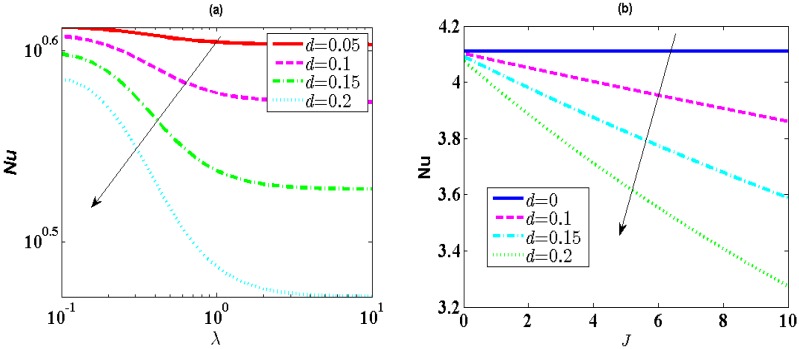
(**a**) The influence of *d* (0.05, 0.10, 0.15, 0.20) on *Nu* with *λ* (10^-1^–10) when *K_λ_* = 1, *R* = 1, *α* = 1, *u_r_* = 0.1, *J* = 5, *Br* = 0.01. (**b**) Influence of *d* (0, 0.10, 0.15, 0.20) on *Nu* with *J* (0–10) when *λ* = 0.25, *K_λ_* = 1, *R* = 1, *α* = 1, *u_r_* = 0.1, *Br* = 0.01.

**Figure 6 micromachines-10-00034-f006:**
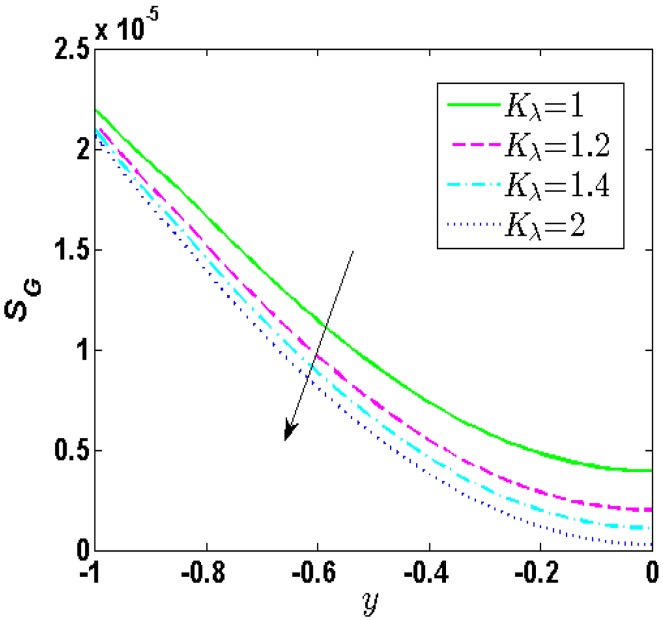
Distribution of entropy generation when *λ* = 0.25, *d* = 0.15, *R* = 1, *α* = 1, *u_r_* = 1, *Br* = 0.02, *J* = 5, *Θ* = 1000.

**Figure 7 micromachines-10-00034-f007:**
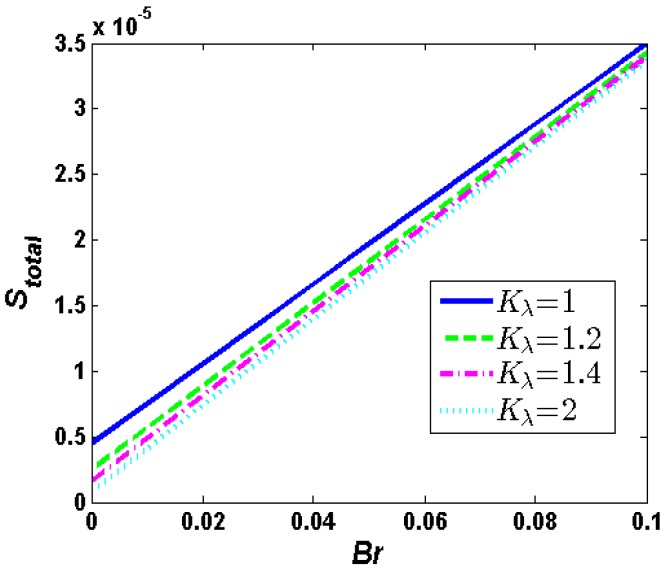
Distribution of total entropy generation when *λ* = 0.25, *d* = 0.15, *R* = 1, *α* = 1, *u_r_* = 1, *J* = 5, *Θ* = 1000.

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
