# Peer review of "Heat Transport of Electrokinetic Flow in Slit Soft Nanochannels"

_micromachines, 2019, doi:10.3390/mi10010034_

Round 1

Reviewer 1 Report

The authors studied the thermal transport properties of fluids through
soft microchannels, giving an analytical solution which is verified by
the finite difference method.  The obtained results are covienent for
analyzing the thermal transport properties of the fluids through soft
microchannels.  I, therefore, recommend the paper for publication in
Micromachines if the following minor points are amended by the
authors.

Comments:

1) English should be improved.

2) The abstract should concisely summarize the main points of the
study; The present abstract is rather lengthy.

Reviewer 2 Report

Please see attached report.

Reviewer 3 Report

The work presented in this manuscript focuses on numerical simulation of heat transport of electrokinetic flow in soft microchannels. This work is highly derivative from an already published work by Chanda et al, Soft Matter, 2014. 

Additionally, there are a number of issues (as follows) that need to addressed.

From the entire manuscript, it is not clear why this work is significant and what is the practical application of this work. It is important for the authors to describe which aspect of real world problems can their work affect or solve.

In the first line of the introduction, authors talk about ink-jet printing as an example of microfluidic devices. It is a method of fabrication, not an example of the devices themselves. Authors are requested to amend the sentences accordingly.

Page 2, line 44, what do the authors mean by 'general microchannels'? This term is used very ambiguously. Please explain or re-define.

Please provide examples of industrial or other applications where heat transfer rates in 'microchannels' is needed and significant.

Why do the authors think that soft microchannels provide a good model to study heat transfer?

In the entire manuscript, there are a number of places where the paramters in the equations have not been defined. The definition of parameters may be obvious to the authors, but it is a good practice to define them nonetheless.

Authors seem to acquire their permissible ranges of the relevant parameters from the paper by Chanda et al. who has worked on nanochannels. Please provide the reason for using the similar ranges even though the dimensions of the channels differ in orders of magnitude.

Round 2

Reviewer 1 Report

The authors studied the thermal transport properties of fluids through
soft microchannels, giving an analytical solution which is verified by
the finite difference method.  The obtained results are convenient for
analyzing the thermal transport properties of the fluids through soft
microchannels.  In addition, the authors satisfactorily revised the
manuscript along the line suggested in the previous report.  
I, therefore, recommend the paper for publication in Micromachines in
the present form.

Author Response

Thank you for your review results.

Reviewer 2 Report

The authors have significantly improved the presentation of the manuscript in response to the questions of all referees. I still have a few lingering questions concerning their derivations. In addition, the English of the manuscript still needs moderate improvement, especially in the Introduction. Accordingly, I recommend publication after moderate revision, provided these questions are answered:

- Please explicitly define the bulk temperature Tm*(x*). At first, I thought it was the temperature T* at the wall, but then I saw that T* = Tw* at y* = -h* (Eq. 5b). Is Tm*(x*) temperature averaged over the channel cross-section?

- Is the application of Newton's Law of Cooling an assumption only valid for sufficiently small temperature differences? Relatedly, does qw* = h* [Tw*(x*) - Tm*(x*)] define the heat transfer coefficient h*?

- In the derivation of Eq. 6 through integration over a control volume, what happened to k*? I understand all terms except the first on the RHS.

- On line 190, p. 6, the authors say that \Theta is a constant due to constant wall heat flux and wall temperature. However, the wall temperature Tw* varies in the x* direction.

Please improve the English in the introduction. There are numerous grammatical mistakes. For instance, the first line would be more grammatical as something like: "In recent years, microfluidic devices, including MEMS, have been widely used in biology, chemistry, medicine, and engineering for chemical separation and thermal management of microelectronic systems." On p. 1, line 23, I suggest changing "equipment themselves" to "these devices".  On p. 1, line 34, "There, it will be induced" could be changed to "Finally, this induces", and "called as" should be "called". On p. 1, line 35, eliminate "finally". This is only on the first page -- please improve the first two paragraphs of the second page.

Reviewer 3 Report

Authors have appropriately addressed the comments.        

Author Response

Thank you for your review results.

Round 3

Reviewer 2 Report

I thank the authors for addressing all questions and comments. The revised manuscript is suitable for publication.